# Jet Ventilation Reduces Coronary Sinus Movement in Patients Undergoing Atrial Fibrillation Ablation: An Observational Crossover Study

**DOI:** 10.3390/jpm13020186

**Published:** 2023-01-20

**Authors:** Cécile Maeyens, Pierre Nokerman, Ruben Casado-Arroyo, Juan-Pablo Abugattas De Torres, Brenton Alexander, Edgard Engelman, Denis Schmartz, Turgay Tuna

**Affiliations:** 1Department of Anesthesiology, H.U.B.—Hôpital Erasme, Université Libre de Bruxelles, 1070 Brussels, Belgium; 2Department of Cardiology, H.U.B.—Hôpital Erasme, Université Libre de Bruxelles, 1070 Brussels, Belgium; 3School of Medicine, University of California, La Jolla, San Diego, CA 92093, USA; 4EW Data Analysis, 1180 Brussels, Belgium

**Keywords:** high-frequency jet ventilation, intermittent positive pressure ventilation, atrial fibrillation ablation

## Abstract

Background: One of the reasons that high-frequency jet ventilation (HFJV) is used is due to the near immobility of thoracic structures. However, no study has quantified the movements of cardiac structures during HFJV compared with normal mechanical ventilation. Methods: After ethical approval and written informed consent, we included 21 patients scheduled for atrial fibrillation ablation in this prospective crossover study. Each patient was ventilated with both normal mechanical ventilation and HFJV. During each ventilation mode, displacements of the cardiac structure were measured by the EnSite Precision mapping system using a catheter placed in the coronary sinus. Results: The median [Q1–Q4] displacement was 2.0 [0.6–2.8] mm during HFJV and 10.5 [9.3–13.0] mm during conventional ventilation (*p* < 0.000001). Conclusion: This study quantifies the minimal movement of cardiac structures during HFJV compared to standard mechanical ventilation.

## 1. Introduction

With a worldwide prevalence of 43.6 million in 2016, atrial fibrillation (AFib) is an increasingly frequent pathology that also creates a significant impact on healthcare costs [1,2,3]. The catheter ablation procedure for the treatment of AFib was first introduced during the 1990s, and when compared with treatment using antiarrhythmic drugs, it improves quality of life by better restoring sinus rhythm [4,5] with a lower recurrence rate [6].

While initially these ablation procedures were performed under conscious sedation, they are now most often performed under general anesthesia [5]. This approach allows for a comfortable and motionless patient [7,8] throughout a potentially long procedure. General anesthesia has been shown to be associated with better stability of the ablation catheter, a better contact force with tissues, and shorter fluoroscopic and procedural times [8,9].

Unfortunately, the movement of the chest and its contents during mechanical ventilation may interfere with the procedure. This has led to the use of high-frequency jet ventilation (HFJV) by some practitioners during ablation procedures [7,10]. The stability of the mapping and ablation catheter, linked to the near immobility of the thoracic structures during HFJV, allows a reduction in operating and fluoroscopic times, fewer recurrences and improves patient outcomes [7,8,11,12,13,14].

However, despite these advantages, HFJV is currently only used in 14% of ablations in the United States and Canada and 4% elsewhere [5]. This type of ventilation requires trained operators, the appropriate equipment and close monitoring of gas exchange.

Outside of atrial ablation procedures, HFJV is also proposed for use during radiofrequency or cryoablation of hepatic, renal and pulmonary tumors [15,16,17,18]. It has also been used for shock wave lithotripsy of kidney stones [19]. Various authors argue that the use of HFJV limits thoracic, diaphragmatic and intraabdominal organ movements [15,16,17,18,19]. This decreased movement is hypothesized to lead to better target contact, lower fluoroscopic time and lower procedure times, with potentially better outcomes. Denys et al. demonstrated the minimal movement of hepatic, renal and pulmonary tumors during HFJV using CT scan measurements. For pulmonary vein isolation, HFJV has also been shown to be associated with optimized catheter contact and better outcomes [7,12,13,14]. The proposed mechanism is likely at least partially related to the lower tidal volumes for HFJV when compared to normal mechanical ventilation. During HFJV, the total tidal volume consists of both the volume delivered by jet ventilation and the volume added by the Venturi effect [20]. This volume has been shown to be lower than the tidal volume during normal mechanical ventilation [21,22].

To our knowledge, no study has quantified this near immobility of cardiac structures observed during HFJV. As such, the primary objective of our study is to measure the displacement of the heart and, in particular, that of the coronary sinus during conventional mechanical ventilation (IPPV) versus HFJV in order to quantify this proposed decreased movement. This was performed using the EnSite Precision™ cardiac mapping system (Abbott Cardiovascular, Plymouth MN, USA) connected to the mapping and ablation catheter placed in the coronary sinus before the start of the ablation procedure.

We hypothesize that the use of HFJV compared to mechanical ventilation will cause less displacement of heart structures, measured as the movement of the Inquiry catheter (Abbott Cardiovascular, Plymouth MN, USA) within the coronary sinus.

## 2. Materials and Methods

This single-center study was approved by the Ethics Committee of Erasme University Hospital on 9 May 2019 (Chairperson Pr JM Boeynaems, reference CCB B406201939759). After written informed consent, 21 patients scheduled for atrial fibrillation ablation under general anesthesia were included in this prospective, interventional study. Inclusion criteria were: Patients over 18 years scheduled for ablation of atrial fibrillation, body mass Index (BMI) ≤ 30 kg/m^2^ and American Society of Anesthesiologists (ASA) score 1–3. Exclusion criteria were: Refusal to participate in the study, pre-existing pulmonary disease, and heart failure defined as a left ventricular ejection fraction <30%.

Patients were equipped with two peripheral intravenous lines, standard monitoring consisting of pulse oximetry, ECG, non-invasive blood pressure monitoring as well as a radial arterial catheter inserted before anesthesia induction. Anesthesia was induced by a target-controlled infusion of propofol and remifentanil. Muscle relaxation was obtained using bolus rocuronium injections adjusted to keep train of four ratio (TOF) <2, measured using a TOF-Watch monitor (Organon, Swords, Ireland). A transesophageal echo (TEE) probe was used for the transeptal puncture and to assess volume status. Mean arterial pressure was maintained at >65 mm Hg using a provider-adjusted norepinephrine infusion.

After the induction of anesthesia, a Wei jet endotracheal tube (Well Lead Medical Co., Ltd., Guangzhou, China) was inserted. This tube allows either conventional mechanical ventilation or HFJV. Positive pressure mechanical ventilation was started with a tidal volume of 5 to 8.5 mL/kg of ideal body weight, a frequency of 10 to 14 breaths per minute adapted to obtain an end-tidal CO_2_ between 35 and 45 mm Hg, an inspired oxygen fraction adapted to obtain a SpO_2_ greater than 95%, and a positive end-expiratory pressure (PEEP) of 0–6 cm H_2_O. HFJV was delivered by a Monsoon III jet ventilator (Acutronic Medical Systems AG, Hirzel, Switzerland). Initial conditions of HFJV were a driving pressure of 1 to 2 bars, an inspired oxygen fraction of 40 to 50%, an injection frequency of 100 to 150 breaths per minute and an inspiratory time to expiratory time (I/E) ratio of 30% to 35%. The gas humidification system of the jet ventilator was used during the procedure.

A blood gas measurement was performed every 30 min [23]; the driving pressure and the I/E ratio were then adjusted in order to maintain an arterial pH between 7.35 and 7.45 and a PaCO_2_ between 35 and 45 mmHg.

The inspired oxygen fraction was set between 40 and 50% and increased if necessary in order to maintain arterial oxygen saturation greater than or equal to 95% of SpO_2_.

### 2.1. Ablation Procedure

Atrial fibrillation ablation was done using radiofrequency (Tacticath, Abbott Cardiovascular, Plymouth MN, USA) subsequent to electrophysiological mapping. The mapping catheter (HD grid, Abbott Cardiovascular, Plymouth MN, USA) and a coronary sinus catheter (Abbott Inquiry, Abbott Cardiovascular, Plymouth MN, USA) were inserted through femoral venous access. The mapping and the coronary sinus catheter were connected to the Abbott EnSite Precision cardiac mapping system (Abbott Cardiovascular, Plymouth MN, USA). After a stabilization period, the coronary sinus catheter was locked into position. This reference point was used to measure cardiac displacement during the study protocol. The coronary sinus catheter, together with the Abbott EnSite Precision system, is able to measure a minimum displacement of 0.34 mm [24,25].

The Abbott Inquiry catheter has multiple electrodes to measure its position. These electrodes are placed in a specific pattern: one electrode, 2 mm spacing, one electrode, 5 mm spacing, and this pattern is repeated five times. The electrodes can be seen in Figure 1 as black lines on the catheter. A region of interest, seen in blue in Figure 1, is defined as a three-dimensional volume around the catheter, and the software measures all movement in any direction inside this volume. The software needs between 30 to 45 s to acquire a stable signal to make a correct measurement. We measured the maximum catheter displacement in any direction after this stabilization period. During mechanical ventilation, this represents the movement between the end-inspiratory and end-expiratory positions. During HFJV, there is only an oscillation of the catheter induced by the jet ventilation.

The primary endpoint was observed mobility, which was calculated as the displacement of the heart during conventional mechanical ventilation (IPPV) versus HFJV. This measurement was performed by the EnSite Precision™ cardiac mapping system connected to the Inquiry catheter placed in the coronary sinus before the start of the ablation procedure.

The secondary objectives were to assess the quality of ventilation by monitoring PaO_2_ and PaCO_2_ during HFJV.

### 2.2. Statistics

Based on observational data in our institution, we calculated a required sample size of 15 patients to show a significant difference between the 2 ventilation modes. Assuming a 25% dropout ratio, we decided to include at least 21 patients. Data are presented as median and 25th, 75th percentiles.

The primary endpoint was compared during conventional ventilation and jet ventilation using Student’s t-test for paired data.

Ventilation parameters were compared using an analysis of variance for repeated measures with mixed models. If a *p*-value <0.05 was obtained for the analysis of variance, a Bonferroni test was used for individual comparison of time points.

All statistical tests were two-tailed, and a *p*-value < 0.05 was considered statistically significant. All analyses were performed using the NCSS 20.0.3 statistical package (NCSS, LLC; Kaysville, UT, USA).

## 3. Results

From 22 May 2019 to 11 March 2020, 73 patients scheduled for AFib ablation were screened. Of these, 52 patients were excluded: 14 patients refused to participate, 27 patients did not meet the inclusion criteria or had an exclusion criterion (3 pulmonary arterial hypertension, 6 pulmonary diseases, 5 heart failures with an ejection fraction less than 30%, 13 BMI greater than 30 kg/m^2^) and 11 patients had a logistical problem (e.g., HFJV ventilation not available).

Patient characteristics are shown in Table 1.

### Coronary Sinus Movement

The primary endpoint, coronary sinus movement during ventilation, was decreased during jet ventilation compared with conventional ventilation.

The median [Q1–Q4] displacement was 2.0 [0.6–2.8] mm during HFJV and 10.5 [9.3–13.0] mm during conventional ventilation (*p* < 0.000001).

The three images (Figure 1) illustrate this displacement during conventional ventilation (15 mm) and HFJV (4 mm) in patient n ° 1.

Table 2 shows the evolution of oxygenation and carbon dioxide elimination.

In both groups, 86% of patients needed a norepinephrine infusion to maintain arterial blood pressure above 65 mm Hg.

## 4. Discussion

In this study, we have demonstrated that during HFJV, as compared to classic mechanical ventilation, coronary sinus movement is significantly decreased. Many authors praise HFJV for a motionless field, but this has rarely been quantified. Denys et al. measured the movement of renal, hepatic and pulmonary tumors during radiofrequency treatment by CT scan [18]. They found movements of less than 0.3 mm in the *x*- and *y*-axis; in the *z*-axis, the movement was less than the slice thickness of 3.75 mm. Goode et al. found that for AFib ablation, HFJV leads to a shorter procedural time, more stable conditions in the left atrium and fewer ablation attempts [7]. They explain their findings as secondary to a more stable field with less movement during HFJV. Sivasambu et al. reported a better outcome for AFib ablation using HFJV as the time to atrial fibrillation recurrence was greater after HFJV [12].

Our study is in concordance with these and other studies from the literature, citing a stable field as the main reason for the improved quality of AFib ablation using HFJV [10]. One strength of our study is the use of the cardiac mapping system to measure coronary sinus displacement. The EnSight system is capable of measuring very small movements, thus allowing for a true measurement of the movement of cardiac structures. Additionally, every patient was their own control as measurements were done in every patient during mechanical ventilation and during HFJV.

Oxygenation and paCO_2_ remained stable throughout the procedure, with no difference between HFJV and traditional mechanical ventilation. PaO_2_ and SaO_2_ were similar between the two ventilation modes without the need to increase the inspired oxygen fraction. Carlon et al. also demonstrated that at a similar PEEP level, HFJV and conventional mechanical ventilation resulted in similar arterial oxygenation [26]. PaCO_2_ is dependent on the driving pressure and the I/E ratio. The average driving pressure in our study did not exceed 2 bar, staying within the limits of safety [21,22]. The need to do arterial blood gas analysis to measure paCO_2_ is a disadvantage of HFJV [27]; however, an arterial line is also useful for repeated activated clotting time measurements necessary in atrial fibrillation ablation. As atrial fibrillation ablation becomes a common procedure, it should be noted that many institutions no longer perform routinely invasive arterial pressure management. In this context, the difficulty in measuring paCO_2_ during HFJV represents a challenge. We must emphasize that our study was not powered to evaluate a difference within the secondary outcomes, so these findings are hypothesis-generating only.

When discussing limitations, one might argue that the coronary sinus catheter is moving in the much larger coronary sinus and, thus, is not a correct surrogate for the movement of cardiac structures. However, in this study, after placement in the coronary sinus, the Inquiry catheter was locked in this position, making an isolated movement of the catheter highly improbable. As discussed above, there have been studies linking HFJV with improved atrial fibrillation ablation outcomes [12,13]. As our study quantifies the minimal movement of cardiac structures during HFJV, it is tempting to link the absence of movement to better contact between the cardiac structures and the ablation catheter and, thus, better results. However, this hypothesis must be proven in prospective randomized trials before advocating for the more widespread use of HFJV [28]. There may be potential unforeseen issues that may only present in randomized studies. For instance, Simvasambu et al. found that norepinephrine infusion was required in a higher proportion of patients during HFJV [12]. In our study, the same proportion of patients (86%) needed norepinephrine infusions to keep mean arterial pressure above 65 mm Hg during mechanical ventilation as well as during HFJV, although our crossover design is slightly different. Our study was designed as an observational, pragmatic study, the sequence of ventilation modes was not randomized, and patients were ventilated according to standard practice. Finally, we limited our study to patients with a BMI ≤ 30 kg/m2, and as Hu et al. demonstrated, HFJV may be more challenging in obese patients [29].

## 5. Conclusions

Cardiac displacement during HFJV is 5 times less than during conventional mechanical ventilation. This study quantifies the minimal movement of cardiac structures in patients undergoing atrial fibrillation ablation ventilated using HFJV. Whether this immobility leads to a better outcome must be confirmed in further prospective trials.

## Figures and Tables

**Figure 1 jpm-13-00186-f001:**
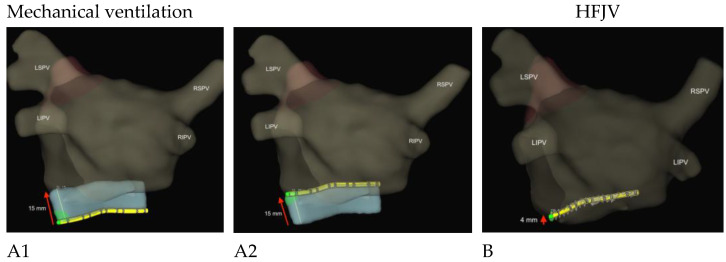
Images of one patient recorded from the Abbott EnSite Precision Cardiac mapping display system, showing the movement of the Inquiry catheter (in yellow) in the coronary sinus. (**A1**,**A2**): during mechanical ventilation, maximum displacement is 15 mm. (**B**): during HFJV, maximum displacement is 4 mm. The blue area shows the amplitude of the catheter movement inside the coronary sinus. The red arrows indicate the range of motion in mm.

**Table 1 jpm-13-00186-t001:** Demographic data.

Age (y)	59.0 [55.0–70.0]
Gender (M/F)	18/3
Weight (kg)	84.0 [71.0–92.0]
Height (m)	1.78 [1.72–1.82]
BMI (kg/m2)	26.3 [24.0–28.0]
All values are median [quartile1–quartile4]	

**Table 2 jpm-13-00186-t002:** Respiratory parameters (Values are median [Q1–Q4]. After 90 min of HFJV, the number of patients drops as the procedure is finished for some patients.)

	PaO_2_ (mm Hg)	SaO_2_ (mm Hg)	PaCO_2_ (mm Hg)	*p*-Value
End of mechanical ventilation	165 [144–289]	99 [98–99]	41 [40–44]	NS
HFJV Start	160 [147–261]	99 [98–99]	42 [40–44]	NS
HFJV 30 min	154 [119–205]	99 [98–99]	46 [37–52]	NS
HFJV 60 min	170 [128–209]	99 [98–98]	40 [37–50]	NS
HFJV 90 min	170 [137–193]	99 [98–99]	41 [38–46]	NS
HFJV 120 min	168 [145–190]	99 [99–99]	41 [34–47]	NS
HFJV 150 min	157 [143–173]	99 [98–99]	42 [37–44]	NS
HFJV 180 min	174 [162–186]	99 [98–99]	41 [37–45]	NS
HFJV 240 min	194	98	34	NS

## Data Availability

Data are available from the corresponding author upon reasonable request.

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
