# Peer review of "Jet Ventilation Reduces Coronary Sinus Movement in Patients Undergoing Atrial Fibrillation Ablation: An Observational Crossover Study"

_jpm, 2023, doi:10.3390/jpm13020186_

Round 1
Reviewer 1 Report
Thank you to let me the opportunity to review this manuscript
The point of this study is to validate one hypothesis of the better conditions of operators during the jet ventilation by showing that the mobility of the heart and vessels is decreased when compared to a conventional ventilation mode, ie the procedure may be easier when the motion of cardiac structures are decreased.
However, even if this manuscript may not change the practices of anesthesiologists unless actively asked by the cardiologist operators, I would much better precise the method of the measurement. All the aim of this study is based on this distance, so I engaged the authors to much better define the procedure, the timing, the exact measured point (fig 1 depicts a yellow line wherein the distance from the baseline may vary according to each point of the line, so what was the exact point chosen to measure this motion ?…)
…. As explained by the authors in the discussion section, presenting results according 3 dimensions (x, y, and z) may be clearer, but the opportunity of the mapping may allow one dimension? So please clarify again the method of measurement.
So they are more questions than answers despite on the final result in millimeters. Wherein this result (a distance) is clear for the reader, the process to obtain this easy to understand value is however not clear at all.
Is the distance observed in the control group linked to the end insufflation (maximal distance) and to the end exsufflation ? Please provide again the exact timing of the measurement according to the phase of the ventilation for the control group.
L119: I don’t understand the measurement process: is it the distance between the extremity of the coronary sinus catheter and the extremity of the mapping catheter?
In the material and methods section, the timing of the protocol should be better explained: how many measurements were taken? how long? When exactly? Please precise if the measurement was a mean of several values (how much) and if so, what was the intra individual variation of the measurement (using for instance the coefficient of variation or another statistical method exploring the variation of an intraindividual measurement…)
We understand better with the figure, but please modify the sentence, to understand clearly the distance measured.
It is surprising that authors did not randomize this cross over study: the effect of a ventilation mode may have affected the lung volume for the next ventilation mode. The time effect would also have been limited. Finally, having two periods of measurement for one ventilation mode (ie 4 time plots by alternating the 2 ventilation modes) may have allowed to obtain stronger values.
Another point is also surprising, why authors did not ventilate all the patients with the same tidal volume (for instance 6 ml/kg of ideal body weight)? Ranging from 5 to 8 ml/kg deserves the control group. In the same way, PEP may have been kept constant.
For non-specialized anesthesiologists, using jet ventilation needs invasive arterial gas? If so, authors should add this disadvantage of using the jet ventilation during such a long time.
Author Response
Reviewer 1
We thank you for your review. Please find enclosed our point-to-point answers, as well as the revised text. (in blue)
Comments and Suggestions for Authors
Thank you to let me the opportunity to review this manuscript
The point of this study is to validate one hypothesis of the better conditions of operators during the jet ventilation by showing that the mobility of the heart and vessels is decreased when compared to a conventional ventilation mode, ie the procedure may be easier when the motion of cardiac structures are decreased.
However, even if this manuscript may not change the practices of anesthesiologists unless actively asked by the cardiologist operators, I would much better precise the method of the measurement. All the aim of this study is based on this distance, so I engaged the authors to much better define the procedure, the timing, the exact measured point (fig 1 depicts a yellow line wherein the distance from the baseline may vary according to each point of the line, so what was the exact point chosen to measure this motion ?...)
.... As explained by the authors in the discussion section, presenting results according 3 dimensions (x, y, and z) may be clearer, but the opportunity of the mapping may allow one dimension? So please clarify again the method of measurement.
So they are more questions than answers despite on the final result in millimeters. Wherein this result (a distance) is clear for the reader, the process to obtain this easy to understand value is however not clear at all.
Is the distance observed in the control group linked to the end insufflation (maximal distance) and to the end exsufflation ? Please provide again the exact timing of the measurement according to the phase of the ventilation for the control group.
L119: I don’t understand the measurement process: is it the distance between the extremity of the coronary sinus catheter and the extremity of the mapping catheter?
In the material and methods section, the timing of the protocol should be better explained: how many measurements were taken? how long? When exactly? Please precise if the measurement was a mean of several values (how much) and if so, what was the intra individual variation of the measurement (using for instance the coefficient of variation or another statistical method exploring the variation of an intraindividual measurement...)
We understand better with the figure, but please modify the sentence, to understand clearly the distance measured.
We agree with the reviewer that methods section needs to be improved to make our measurements clearer to the reader who may be unfamiliar with this procedure. The Abbott Inquiry catheter (Abbott Cardiovascular, Plymouth, MN, USA) has multiple electrodes to measure its position. These electrodes are placed in a specific pattern: one electrode, 2 mm spacing one electrode, 5 mm spacing and this pattern is repeated 5 times. On the figures these electrodes are represented as black lines on the catheter. In order to measure the displacement, a three-dimensional volume is defined as a region of interest around the catheter. The software is able to measure the movement of the catheter in any direction inside this volume. This volume is represented on the figure as a blue area. The software needs between 30 and 45 seconds to acquire a stable signal to make a correct measurement. In conventional ventilation, this period represents around 5-8 ventilation cycles and in high frequency ventilation the measurements were taken after this stabilization period. We measured the maximum displacement on any electrodes in any direction. During mechanical ventilation, this represents the distance between end-inspiratory and end-expiratory position. During high frequency jet ventilation, the movements were minimal during the oscillations induced by the jet ventilation.
We inserted a new paragraph in the method section to explain our measurements in more details. The manuscript (materials and methods section) was changed to:
“The Abbott Inquiry catheter has multiple electrodes to measure its position. These electrodes are placed in a specific pattern: one electrode, 2 mm spacing, one electrode, 5 mm spacing and this pattern is repeated five times. The electrodes can be seen on figure 1 as black lines on the catheter. A region of interest, seen in blue on figure 1, is defined as a three-dimensional volume around the catheter and the software measure all movement in any direction inside this volume. The software needs between 30 to 45 seconds to acquire a stable signal to make a correct measurement. We measured the maximum catheter displacement in any direction after this stabilization period. During mechanical ventilation this represents the movement between end-inspiratory and end-expiratory position. During HFJV there is only an oscillation of the catheter induced by the jet ventilation.”
It is surprising that authors did not randomize this cross over study: the effect of a ventilation mode may have affected the lung volume for the next ventilation mode. The time effect would also have been limited. Finally, having two periods of measurement for one ventilation mode (ie 4 time plots by alternating the 2 ventilation modes) may have allowed to obtain stronger values.
In our institution, cardiologists ask for high frequency jet ventilation for all ablation procedures. So, the standard procedure is to ventilate the patient by conventional mechanical ventilation after induction, the time to insert eventually necessary catheters and to do a thorough TEE exam. After this, the patient is switched to high frequency jet ventilation for the ablation procedure. As we designed the study as an observational study, in order not to further prolong an already long procedure, we decided not to randomize the ventilation sequence. We agree that this represents a shortcoming of our study.
Another point is also surprising, why authors did not ventilate all the patients with the same tidal volume (for instance 6 ml/kg of ideal body weight)? Ranging from 5 to 8 ml/kg deserves the control group. In the same way, PEP may have been kept constant.
Again, the study was designed as an observational, pragmatic trial; so, we did not impose very strict ventilation conditions. The parameters used are in the range used in clinical practice. We agree with the reviewer that this may introduce more variations, however, our measures are closer to the everyday practice. We added this to as a shortcoming to our manuscript, which was changed (discussion) as follows:
“Our study was designed as an observational, pragmatic study, the sequence of ventilation modes was not randomized, and patients were ventilated according to standard practice.”
For non-specialized anesthesiologists, using jet ventilation needs invasive arterial gas? If so, authors should add this disadvantage of using the jet ventilation during such a long time.
During jet ventilation, end-tidal CO2 cannot be measured, so it is recommended to obtain intermittent blood gas samples to maintain normocapnia. (S. Babapoor-Farrokhran et al, J Innov Cardiac Rhythm Manage, 2021; 12(7):4590-4593) However, during atrial fibrillation ablation activated coagulation (ACT) is measured 1 or 2 times per hour as anticoagulation by heparin is mandatory. So, an arterial catheter serves both purposes. We added arterial blood gas sampling to the disadvantages of high frequency jet ventilation. The manuscript (discussion) was changed to:
“The need to do arterial blood gas analysis to measure paCO2 is a disadvantage of HFJV [27]; however, an arterial line is also useful for repeated activated clotting time measurements necessary in atrial fibrillation ablation.”
Reviewer 2 Report
The authors cite many articles in which apparently HFJV was already used for afib treatement , I think they should justify in which circumstances this new work is different than previously published studies.
I know that this was not objective of the study , but did the HFJV reduced the time of the procedure , or the time of fluoroscopy? at least in coùmparison to a historical group?
Why the quality of ventilation was an objective , this has beel already assessed in the many papers avout HFJV , this should be explained or removed
During HFJV there is a tendency for the rise of pco2; although this was not an issue in this study , this shoul be discussed as a rise in PCO2 could yiel arrythmia
The conclusion should state the result of the study not the objective
Author Response
Reviewer 2
We thank you for your review. Please find enclosed our point-to-point answers, as well as the revised text. (in blue)
Comments and Suggestions for Authors
Submission Date Date of this review
The authors cite many articles in which apparently HFJV was already used for afib treatement , I think they should justify in which circumstances this new work is different than previously published studies.
We agree with the reviewer that many studies have compared high frequency ventilation to conventional mechanical ventilation for atrial fibrillation ablation. Many of these studies argues that the minimal movement during high frequency jet ventilation may be responsible for a better outcome. However, to our knowledge, no study has made an objective measurement of the movement during jet ventilation compared to mechanical ventilation. So, the primary objective of our study was to measure the displacement of thoracic structures during both modes of ventilation.
I know that this was not objective of the study , but did the HFJV reduced the time of the procedure , or the time of fluoroscopy? at least in coùmparison to a historical group?
In our institution, cardiologists ask for high frequency jet ventilation for atrial fibrillation ablation procedures, so high frequency jet ventilation is “standard of care” and we do not have a historical group for comparison.
Why the quality of ventilation was an objective , this has beel already assessed in the many papers avout HFJV , this should be explained or removed
The primary objective was to measure thoracic structure displacement. We measured quality of ventilation by measuring PaO2 and paCO2 as during jet ventilation, these parameters are more difficult to maintain in the normal range than during mechanical ventilation.
During HFJV there is a tendency for the rise of pco2; although this was not an issue in this study, this shoul be discussed as a rise in PCO2 could yiel arrythmia
This was the raison why measured paCO2 and adapted driving pressure and I/E ratio during as recommended during high frequency jet ventilation. (REF idem)
The conclusion should state the result of the study not the objective
We changed our conclusion to be more specific. The manuscript (conclusion) was changed to:
“Cardiac displacement during HFJV is 5 times less than during conventional mechanical ventilation.”